# Incorporating Machine Learning into Established Bioinformatics Frameworks

**DOI:** 10.3390/ijms22062903

**Published:** 2021-03-12

**Authors:** Noam Auslander, Ayal B. Gussow, Eugene V. Koonin

**Affiliations:** National Center for Biotechnology Information, National Library of Medicine, National Institutes of Health, Bethesda, MD 20894, USA; ayal.gussow@nih.gov

**Keywords:** machine learning, deep learning, bioinformatics methods, phylogenetics

## Abstract

The exponential growth of biomedical data in recent years has urged the application of numerous machine learning techniques to address emerging problems in biology and clinical research. By enabling the automatic feature extraction, selection, and generation of predictive models, these methods can be used to efficiently study complex biological systems. Machine learning techniques are frequently integrated with bioinformatic methods, as well as curated databases and biological networks, to enhance training and validation, identify the best interpretable features, and enable feature and model investigation. Here, we review recently developed methods that incorporate machine learning within the same framework with techniques from molecular evolution, protein structure analysis, systems biology, and disease genomics. We outline the challenges posed for machine learning, and, in particular, deep learning in biomedicine, and suggest unique opportunities for machine learning techniques integrated with established bioinformatics approaches to overcome some of these challenges.

## 1. Introduction

Over the past few decades, the advances in computational resources and computer science, combined with next-generation sequencing and other emerging omics techniques, ushered in a new era of biology, allowing for sophisticated analysis of complex biological data. Bioinformatics is evolving as an integrative field between computer science and biology, that allows the representation, storage, management, analysis and investigation of numerous data types with diverse algorithms and computational tools. The bioinformatics approaches include sequence analysis, comparative genomics, molecular evolution studies and phylogenetics, protein and RNA structure prediction, gene expression and regulation analysis, and biological network analysis, as well as the genetics of human diseases, in particular, cancer, and medical image analysis [1,2,3].

Machine learning (ML) is a field in computer science that studies the use of computers to simulate human learning by exploring patterns in the data and applying self-improvement to continually enhance the performance of learning tasks. ML algorithms can be roughly divided into supervised learning algorithms, which learn to map input example into their respective output, and unsupervised learning algorithms, which identify hidden patterns in unlabeled data. The advances made in machine-learning over the past decade transformed the landscape of data analysis [4,5,6]. In the last few years, ML and particularly deep learning (DL) have become ubiquitous in biology (Figure 1). However, clinical applications have been limited, and follow-up mechanistic investigation of ML-based predictions is often lacking, due to the difficulty in the interpretation of the results obtained with these techniques. To overcome these problems, numerous approaches have been developed to incorporate ML and DL into established bioinformatics frameworks, for training data selection and preparation, identification of informative features, or data integration. Such integrated frameworks exploit the power of ML and DL methods, offering interpretability and reproducibility of the predictions.

In this brief review, we survey recent efforts to integrate ML and DL with established bioinformatic methods, across four areas in computational biology. We discuss the strengths and limitations of these integrated methods for specific applications and propose avenues to address the challenges impeding even broader application of ML techniques in biomedical research.

### 1.1. Integrating Machine-Learning into Molecular Evolution Research

Combining computer science approaches with principles of molecular evolution analysis has revolutionized the field of molecular evolutionary studies. Application of diverse and increasingly advanced computational methods has enabled accurate determination of evolutionary distances between species, reconstruction of evolutionary histories and ancestries, identification conserved genomic regions, functional annotation of genomes, and phylogenetics. In recent years, ML methods have been developed to address the challenges faced by molecular evolution research, in particular, by overcoming the difficulties of analyzing increasingly massive sets of sequence and other omics data. Examples of such applications include the use of autoencoders to impute incomplete data for phylogenetic tree construction [7], application of random forest for phylogenetic model selection [8], harnessing convolutional neural networks (CNNs) to infer tree topologies [9] and tumor phylogeny [10], and utilization of deep reinforcement learning for the construction of robust alignments of many sequences [11].

Evolutionary algorithms and strategies have been the most successful in solving diverse bioinformatic problems, far beyond core phylogenetic and molecular evolution tasks. Indeed, a wide range of computational techniques are founded on evolutionary strategies, including application of population-based analysis, fitness-oriented rules or variation-driven research [12,13]. For instance, genetic algorithms (GA) [14] are a type of search heuristic which is inspired by principles of biological evolution. The GA is widely used in for optimization of multiple criteria and for features selection [15,16,17]. Evolutionary approaches underly effectively all types of biological sequence analysis. Therefore, integrating ML with molecular evolution and phylogenetic methods is essential to uncover robust and biologically relevant patterns and discriminative features. For example, recent methods combined sequence attributes, alignment, and phylogenetic trees with ML for protein sequence analysis and clustering [18,19] for tasks as different as identification of determinants of viral pathogenicity and infectivity [20,21,22], prediction CRISPR-Cas9 cleavage efficiency [23] and detection of anti-CRISPR proteins [24,25].

Although numerous bioinformatics methods continue to rely on sequence alignments, the advent of ML gave rise to a variety of alignment-free methods that allow skipping the alignment step and learning directly from unaligned sequences. Alignment-free methods are especially useful, for example, for the identification of viral sequences in complex sequence datasets, where highly divergent viruses are often difficult to identify with straightforward alignment and sequence comparison. Therefore, alignment-free tools have been developed for viral sequence identification by employing ML techniques such as SVM [26], RNN [27] and CNN [28,29]. Alignment-free methods are also useful for the functional annotation of nucleic acids and proteins, where in some cases function may be inferred from particular domains or motifs that can be detected without complete nucleotide or protein alignment. In cases where sequence profiles are difficult to derive, ML and particularly DL techniques can be trained to rapidly recognize specific domains or motifs, without the need to devise explicit sequence profiles [29,30,31]. Several DL techniques have been employed for the annotation of functional features in nucleotide sequences, typically relying on a large, annotated sequence dataset for training, for example, using deep RNN [32,33] or CNN [34,35]. These applications include identification of promoters [36,37], enhancers [38,39], long noncoding RNAs [40,41,42,43,44], microRNA targets [45,46], and CRISPR arrays [47].

The key challenge in the application of ML to molecular evolution and phylogenetics, where traditional bioinformatic strategies efficiently resolve many substantial problems, is the identification of tasks that have not been yet properly addressed, but involve learnable patterns and features. This challenge stems from the difficulty of estimating the learnability of different problems, but also, from the shortage of labeled datasets of sufficient size for problems that are not easily amenable to standard bioinformatic techniques.

### 1.2. Integrating Machine-Learning with Protein Structure Analysis

In the study of proteins, numerous methods have been developed to process the amino acid sequence, and predict the protein structure, function and post translational modifications, such as phosphorylation and glycosylation, that are crucial to the function of many if not most proteins. ML techniques have been incorporated with traditional proteomic methods to predict and analyze post translational modifications [48,49], including CNN [50], hierarchical and K-means clustering [51,52]. The Musite suite integrated KNN with the search for local sequence similarity to known phosphorylation sites, protein disorder scoring and amino acid frequency calculation to predict general and kinase-specific phosphorylation sites [53]. EnsembleGly developed an ensemble classifier of protein glycosylation site based on curated glycosylated protein database and SVM [54]. More recently, several DL models have been incorporated with other modeling techniques and curated databases for the prediction of phosphorylation sites [50,55], and protein glycosylation [56].

Fundamental computational challenges in the field of protein analysis include prediction of protein structure from sequence, accurate estimation of structural similarity to infer homology and prediction of protein contact maps [57,58]. Solving these problems is crucial for the characterization of protein functions, localization and interactions, and can directly contribute to many research directions, from deciphering evolutionary history [59] to drug discovery [60]. Existing computational methods for protein structure prediction that rely on thermodynamics, molecular mechanics, heuristics, and similarity to previously solved structures have demonstrated varying levels of success [61,62,63]. ML and particularly DL techniques have recently entered this field but have already shown the potential to revolutionize protein structure prediction, inference of homology from structure comparison and estimation of contact maps.

Numerous ML methods have been developed for protein structural prediction, with particular success achieved with deep learning architectures [58,64]. The Critical Assessment of Structure Prediction (CASP), which assesses prediction methods and models [64], recently noted substantial progress in structure modeling by deep learning, in particular, template free modeling (FM), that is, modeling structure without an existing template, as opposed to homology modeling. Numerous deep learning methods now require fewer proteins in the input MSA and have demonstrated increasing success in FM modelling [65,66,67,68,69,70,71], primarily due to more precise prediction of contact maps and inter-residue distances [64]. Some methods are narrower in scope and focus on contact prediction [72,73,74,75]. The strongest predictor for CASP13, the most recent CASP with a published report, was AlphaFold [76,77], a deep learning predictor from DeepMind. The results from CASP14 have not been yet described in detail but are available online [78]. CASP14 was marked with the striking success of AlphaFold2, the next version of AlphaFold, which integrates established sequence search tools into a deep learning framework. AlphaFold2 employs sequence database search to construct multiple sequence alignments (MSA), and extracts MSA-based features that are given as input to a deep residual convolutional neural network [79]. This network architecture eases the training of deep networks by introducing shortcut connections with gating functions, that avail the input of lower layers to higher layer nodes in the network. In CASP14, AlphaFold2 vastly outperformed every other method, both FM and template-based modeling approaches. The results of AlphaFold2 are so impressive that there seems to be a realistic possibility that this computational approach could begin to replace the expensive and time-consuming protein crystallography and even the more efficient cryo-EM. Regardless of whether and when this promise materializes, it is becoming clear that DL has already revolutionized protein structure analysis, and rapid and broad improvements can be expected to occur in the next few years (Figure 2).

### 1.3. Integrating ML into Systems Biology

The rapid growth and diversification of biological data calls for an increasingly wide range of modeling and analysis techniques to be employed in systems biology. With complex omics datasets that are now incessantly accumulating, there is a growing need for techniques that can integrate different data types, incorporate datasets into established biological networks and combine different systems biology approaches to investigate multi-omics datasets. Various ML methods have been developed to utilize multi-dimensional datasets together with biological networks, study complex interactions and model biological systems. ML techniques in network biology can be classified into those that infer the network architecture and those that integrate existing network architectures with biological data measurements [80]. Consequently, some of these techniques also require sophisticated data integration methods to incorporate different data types into a model.

Different ML frameworks have been utilized for the inference of biological networks, such as the gene regulatory network (GRN) in the DREAM5 project [81] which utilizes SIRENE [82], a support vector machines-based approach for regulatory networks utilization. More recently, a transfer learning technique [83], and a single-cell RNA sequencing based ML technique [84] have also been proposed for GRN reconstruction. ML methods also have been employed for the inference of protein-protein interactions (PPI) networks, for example, by utilizing NMF [85], regression [86], PCA [87] and deep neural networks [88]. Such methods include the recently developed signed variational graph auto-encoder [89], a graph representation learning method that incorporated graph structure and sequence information to study PPI networks, PPI_SVM [90], which integrated support vectors machines with domain affinity and frequency tables, and LightGBM-PPI [91], which utilizes elastic net regression models with different protein descriptors for inference of PPI networks. In addition, several DL-based techniques have been proposed for PPI network reconstruction [88,92,93,94,95]. These methods primarily exploit recent advances in deep learning architectures to enhance the prediction of PPI networks [93]. Network inference techniques were additionally developed to advance disease research, and several ML techniques have been developed to identify drug-target interaction networks using drug similarity [96,97], by integrated K-means clustering with network analysis [98], or by integrating different networks and data types [99,100]. Several DL based techniques have been developed to predict drug response based on cell line data [101,102], by integrating genomic profiles [103], or through multi-omics integration [104]. Some methods incorporate chemical properties of compounds with ML to predict their clinical effects [105,106,107] and recently, a cancer network inference technique has been proposed to identify signal linkers which coordinate oncogenic signals between mutated and differentially expressed genes [108].

ML methods have also been incorporated with established network structures to analyze diverse biological datasets. ML techniques have been incorporated with biological networks to predict anti-cancer drug efficacy [109], to model drug response by integrating prior biological knowledge with different biological data types [110], and by computing “network profiles” based on PPI networks [111]. Several strategies have been proposed to employ ML for network-based prediction of drug side effects [112,113,114] and drug combinations [115], for prediction of synergistic drugs [116,117] and drug repositioning [118,119,120]. Several studies have used machine and deep learning techniques to investigate properties of metabolic networks, such as inference of metabolic pathways [121,122], differential metabolic activity [123] and pathway reconstruction [124,125]. A variety of studies have integrated information obtained for different data types using ML methods, including the integration of network and pathway data for the discovery of drug targets [123,126,127], incorporation of a pathway-derived mechanistic model with gene expression to identify new drug targets [128], and inference of the activity of oncogenic pathways in cancer [129,130]. Recent strategies integrate multi-omics datasets with ML techniques to enhance the prediction of pathway dynamics [131] and utilize pathway based multi-omics integration for patient clustering [132].

With the recent increased availability of multiple, powerful omics techniques (that is, genomics, transcriptomics, proteomics, and metabolomics), a key emerging challenge is the integration of different omics platforms. Several methods have been developed for multi-omics integration using machine and deep learning techniques [133], including SVM [134,135], KNN [136,137], NMF [138], PCA [139] and CNN [140], for example, for cancer subtype and survival prediction [141,142,143] and for prediction of drug response [143,144], the paucity of studies systematically comparing different multi-omics integration methods is a serious bottleneck in the advancement of this field. Such systematic comparison was recently performed for a subset of the multi-omics techniques aimed at the prediction of tumor subtype [145]. The lack of standardized techniques and clear recommendation of methods to use for particular applications may lead to inadequate selection of analysis strategy and overfitting [146].

### 1.4. Integrating ML with Genomics and Biomarker Analysis for Disease Research

In recent years, molecular phenotyping using genetic and genomic information has allowed early and accurate prediction and diagnosis of many diseases, and critically improved clinical decision making [147,148]. In disease research, the key challenges are the identification of disease-associated genes and mutations for diagnosis, and prediction of the disease progression and clinical outcome as well as drug response and personalized medicine.

Traditional algorithms for the identification of disease-associated genes and disease-causing mutations mostly rely on analysis of sequence data, which can be limited for rare diseases. In addition, some diseases are caused by epigenetic alterations, and thus are not linked to specific mutations or genetic variation. Therefore, several techniques have been developed to identify genes that are associated with complex diseases by incorporating machine and deep learning methods with different types of data, biological networks and bioinformatic techniques. For example, incorporation of network analysis of differentially expressed genes with ML allowed the prioritization of disease-genes even without disease phenotype information [149], hierarchical clustering analysis to differentially expressed genes revealed genes associated with pulmonary sarcoidosis [150], and integration of non-negative matrix factorization (NMF) with disease semantic information and miRNA functional information uncovered new miRNA-disease association [151]. Other examples include training machine learning classifiers on gene functional similarities inferred with Gene Ontology (GO) resulting in successful identification of genes associated with the Autism Spectrum Disorder [152], and applying ML to features calculated based on protein sequences, allowing inference of the probability of a protein’s involvement in disease, without considering their function or expression [153]. Furthermore, recently developed algorithms allow ML-based visualization of disease relationships, for example, of disease-phenotype similarity and disease relationships with t-SNE [154,155]. In addition, ML has been integrated with PPI networks to infer a phenotype similarity score and rank protein complexes by phenotypes that are linked to human disease [156], to identify topological features of disease-associated proteins [157], and recently, to identify host genes that are associated with infectious diseases [158]. Furthermore, ML algorithms have been employed for the detection and investigation of cancer driver genes, by incorporation of ML with statistical scoring of genomic sequencing [159], pathway-level mutations [160], mutation and gene interaction data [161], and by application of deep convolutional neural networks and random forest for analysis of mutations and gene similarity networks [162,163].

A biomarker is a biological measure that can be used as an indicator of a disease state or response to therapeutic interventions [164,165]. There are three categories of disease biomarkers. First, risk biomarkers are used to identify patients that are at risk of developing a disease. Second, diagnostic biomarkers help detect a disease state and determine the disease category. Third, prognostic biomarkers help predict disease progression, response to treatment and recurrence [166]. Various ML approaches, and in particular, feature selection methods have been applied to discover molecular biomarkers and classify clinical cases. For example, an approach for the discovery of biomarker signatures has been proposed based on a pipeline that applies feature selection through integration of different data types with biological networks [167]. Several machine learning techniques have been developed for biomarker discovery in cancer, by using protein biomarkers to classify cancer states [168], and developing biomarkers for early cancer diagnosis from microarray and gene expression data [169,170,171,172], urine metabolomics [173,174] and multidimensional omics data [175,176,177]. Several methods have been developed that integrate network information with omics data for biomarker discovery [167,175,178], and some methods incorporated prior knowledge into feature selection algorithms for biomarker discovery, such as diseases associated genes [179,180], evolutionary conservation [179,181], pathway information [182,183,184], and by applying network feature selection [185,186]. Recently, ML techniques were proposed to develop biomarkers that match patients to treatments, such as identification of markers that correlate with enhanced drug sensitivity [103,109,187], and treatment recommendations with SVM [188] and RNN [189] (Table 1).

### 1.5. Key Challenges and Future Directions

ML methods including, recently, DL algorithms have become a rapidly growing research area, redefining the state-of-the-art performance for a wide range of fields [4,5]. Given the rapid growth in the availability of biomedical and clinical datasets in the past decades, these techniques can be expected to similarly transform multiple avenues of biomedical research, and indications of their high efficacy are already accumulating. The success of AlphaFold2 that dramatically outperforms all other existing methods for protein structure prediction from amino acid sequences [77] is perhaps the strongest case in point. It appears more than likely that similar efforts will result in breakthroughs in a variety of biomedical fields through the integration of ML with more traditional bioinformatics approaches. However, there are several key obstacles that have to be overcome to enable the development and acceptance of ML solutions to pressing problems in biomedicine. We discuss some of the most substantial challenges and suggest means to overcome them through integration of ML frameworks with prior biological knowledge, databases, and established bioinformatics techniques (Table 2).

**Table 2 ijms-22-02903-t002:** Challenges posed for ML and DL in biomedicine, existing strategies to overcome these challenges and proposed solutions by integrating ML techniques with established bioinformatics approaches.

Problem	Bottleneck	Example Solutions	Potential Integrated ML/DL and Bioinformatics Solutions
Small and dependent datasets	Data availability	Restricting the number of parameters [27,190]	Neural network architectures for small and sparse datasets
		Separating training and test sets by phylogenetic similarity [27]	Methods to evaluate data dependency by protein and sequence similarities
Biological sequence representation	Methodological	NLP with neural networks-based modeling [191,192,193,194]	Incorporating amino acid substitution and codon usage matrices to representation frameworks
			Incorporating conserved domain databases to the training framework
Incorporation of different data types	Methodological	Integration of multi-omics datasets through existing network topologies	
Reproducibility	Acceptance	Documentation and deposition of the processed data [195]	-
		Benchmarking of the processing pipeline and optimized parameters [196]	-
Interpretability	Acceptance	Incorporation of established bioinformatic methods and databases with ML and DL frameworks [128,196]	
		Generation of interpretable DL models [197,198,199]	

A major challenge for the application of ML and particularly DL to biological sequences is the representation of nucleotides or amino acid sequences as a sequence of numbers or vectors. Representation of biological sequences as well as feature extraction methods for genetic, molecular and clinical data are imperative for the subsequent successful application of ML and DL techniques. The leading method developed for biological sequence representation is BioVec [191], which includes GeneVec, a representation of gene sequences, and ProtVec that represents protein sequences. BioVec relies on the Word2Vec algorithm [200], a natural language processing (NLP) technique that employs a neural network-based model, and is applied to n-gram representations of the protein sequence. This approach has been applied to protein family classification and visualization of proteins [191]. More recent methods for distributed representation of biological data operate by learning gene co-expression patterns [192], representation of cancer mutations [193], and representation of residue-level sequences for kinase specific phosphorylation site prediction [194]. These efforts are almost entirely data-driven, and do not make use of the curated databases and bioinformatic tools that are widely employed for the analysis of biological sequences. For example, well established matrices that have been designed to evaluate amino acid substitutions [201] and codon usage [202] could be considered when encoding biological sequences. Furthermore, numerous manually curated conserved domains databases that document functional and structural units of proteins [203] could be integrated into the training and evaluation steps of DL frameworks for protein annotation and functional classification. Incorporation of curated databases and established bioinformatic matrices into sequence representation methods is expected to enhance the training, evaluation and interpretability of DL models.

One consequence of the lack of efficient protein sequence representation is a frequent use of the simplest, assumption-free representation, which is one-hot encoding, where each position in a sequence is represented by a 20-dimensional vector with 19 positions set to 0 and the position identifying a specific amino acid set to 1. Although the one-hot representation can sometimes outperform other scales [204], one-hot encoded protein sequences are sparse, memory-inefficient and high-dimensional [205]. In addition, one-hot encoding lacks the notion of similarity between sequences, and thus, is more appropriate for categorical data with no relationship between the categories [205]. This could be a particularly severe problem when a one-hot representation is given to a convolutional neural network. Most convolutional layers identify spatial patterns in the data, which the one-hot encoding inherently lacks. By using a sparse, one-hot encoded protein sequences, a deep convolutional network can wrongly infer similarity patterns and spatial connections between amino acids, which could be meaningless and could lead to overfitting [206,207]. In addition, a convolution is more likely to capture local and proximal patterns and dismiss long-range patterns [208], which is problematic for any sparse representation, but especially, when long-range dependencies are known or suspected to exist in the data. Therefore, it is crucial to carefully consider the appropriate data representation and neural network architecture for every prediction problem.

Despite the advent of the big data era, for many major challenges in biomedicine, the available data are small, sparse, and highly dependent. This is a major problem for training DL models, which require massive amounts of training data and an independent test set. Biological data, and especially biological sequence databases, tend to include high proportion of duplicate or near-duplicate samples [209], which can seriously bias learning algorithms, especially when duplicates are present between the training and test datasets [210,211,212]. For the training and evaluation of DL algorithms on highly dependent biological data, careful data processing is needed to minimize duplicates and near-duplicates and ensure independence between the training and test sets [27,213]. With the growing availability and appeal of DL frameworks, the issues of sample size and the independence of biological data are frequently ignored, so that large-scale models are trained without data filtering and preparation, and therefore without ever being evaluated on a truly independent test. To overcome these limitations, it is necessary to develop neural network architectures that are specifically designed for small and sparse datasets [27,214,215]. In addition, there is a pressing need for the development of methods that estimate the dependencies between biological samples using existing bioinformatics techniques (such as clustering of nucleic acid and proteins by sequence similarity), with subsequent evaluation of the maximum model size and the number of parameters given the true size of independent samples.

Another important challenge in biomedical applications of ML is the difficulty in incorporating different data types. With the growing availability of multi-omics datasets that combine genomics, transcriptomics, metabolomics and proteomics data, there is a pressing need for systematic evaluation of the strategies for multi-omics integration techniques, and for the assessment and development of learning algorithms that can be applied to integrated datasets. In particular, methods are required for data reduction, visualization, and feature selection that allow a combined view and evaluation of integrated multi-omics datasets. Integration of multi-omics datasets through incorporation of curated network topology can enhance the development of multi-omics ML pipelines, and provide means for feature connection, selection and reduction based on established biological networks.

Reproducibility is another major issue that has been extensively discussed in the context of biomedical applications of ML and other computational techniques [216]. Code sharing and open-source licensing and sufficient documentation and additional recommended practices are crucial factors to allow reproducibility of computational biology methods [195]. In bioinformatics research, poor reproducibility can also be attributed to data processing, where different pipelines can differ even in estimations for the same dataset [217,218,219]. Documentation and deposition of the processed data are imperative, and when possible, benchmarking of the processing pipeline and optimized parameters can substantially increase the reproducibility of ML approaches.

Last but not least, the lack of interpretability is a principal issue impeding the widespread usage and adaptation of ML, and especially DL techniques in bioinformatics research. Investigation of the biological mechanisms underlying the success of predictive models and features is highly desirable for the acceptance and use of these techniques, and particularly for clinical applications. Despite several important efforts to improve interpretability of DL models in biomedicine [197,198,199], model interpretability research in genomic and medicine is highly underdeveloped. Common techniques to address the interpretation of concepts learned by a deep neural network include activation maximization, which identifies input patterns that maximize a desired model response [35,220]; sensitivity analysis or network function decomposition, aimed to explain the network’s decisions and input representation [220,221,222]; and layer-wise backpropagation, which propagates the prediction to highlight the supporting input features [223]. Use of bioinformatic techniques, for example, for input representation, will enhance the interpretation of these analyses by revealing biological implications of the input patterns. Therefore, incorporation of established bioinformatics methods and curated databases into ML frameworks is a powerful way to increase the interpretability of these approaches, enhance their utility and use in biomedicine, and allow for follow-up investigation and derivation of hypotheses.

## 2. Conclusions

Machine learning and deep learning in particular are powerful computational tools that have already revolutionized many domains of research. With the recent expansive growth of genomic, molecular, and clinical data, ML offers unique solutions for the interrogation, analysis, and processing of these data, and for extracting substantial new knowledge on the underlying processes. The ML techniques are especially appealing in computational biology because of their ability to rapidly derive predictive models in the absence of strong assumptions about the underlying mechanisms, which is typical of some of the most pressing challenges in biomedicine. However, this unique ability also imposes serious obstacles for the development and widespread acceptance of the ML and particularly DL methods, impeding the reproducibility and interpretability of predictive models. Researchers in biomedical fields often lack the background and skills to perform or evaluate ML and especially DL analysis, which may lead to erroneous practices and conclusions [224]. The development of ML frameworks for biomedicine requires expertise in biology or clinical research, to comprehend and evaluate the strengths and limitations of intricate biological and clinical data, to be combined with a strong background in data mining and computational techniques.

Incorporation of ML techniques into established bioinformatics and computational biology frameworks has already notably facilitated the development of predictive models and powerful tools in molecular evolution, proteomics, systems biology, and disease genomics. The reliance on bioinformatics frameworks for data processing, training and evaluation of predictive models has been instrumental for the use and acceptance of these techniques in biomedicine, and such integrated approaches present promising solutions for many of the major obstacles for machine learning in biology and medicine.

## Figures and Tables

**Figure 1 ijms-22-02903-f001:**
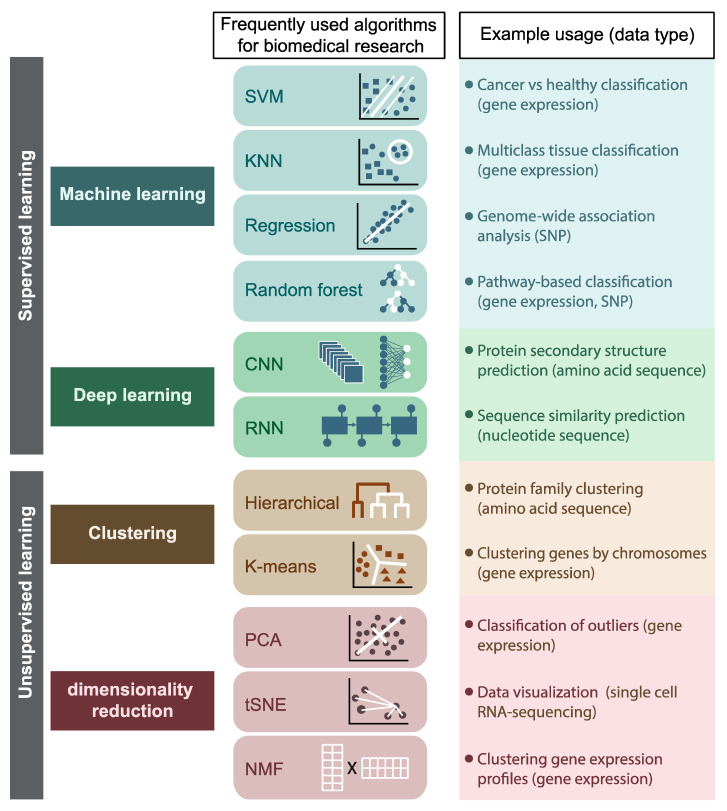
Machine learning algorithms frequently used in bioinformatics research. An example of the usage of each algorithm and the respective input data are indicated on the right. Abbreviations: SVM, support vector machines; KNN, K-nearest neighbors; CNN, convolutional neural networks; RNN, recurrent neural networks; PCA, principal component analysis; t-SNE, t-distributed stochastic neighbor embedding, NMF, non-negative matrix factorization.

**Figure 2 ijms-22-02903-f002:**
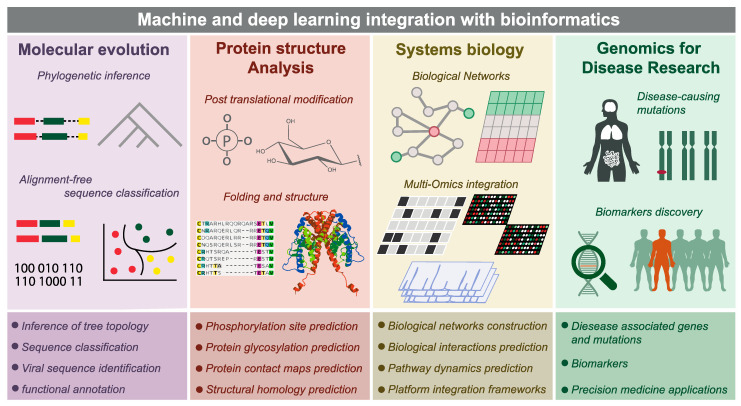
Applications of integrated machine learning techniques with bioinformatics in molecular evolution, protein structure analysis, systems biology, and disease genomics.

**Table 1 ijms-22-02903-t001:** Representative problems and methods addressing them by incorporating machine learning (ML) with bioinformatics tools in four areas.

Bioinformatics Area	Problem Category	Goal	ML Method	Bioinformatic Tools
Molecular evolution	Biological sequence clustering	Protein family prediction	CNN	Clusters of Orthologous Groups (COGs) and G protein-coupled receptor (GPCR) dataset [30]
		Protein function prediction	deep RNN	BLAST and HMMER search [32]
		Anti-CRISPR proteins identification	Random forest	MSA and PSI-BLAST [24]
			EXtreme Gradient Boosting	K-mer based clustering (CD-HIT), BLAST [25]
		Viral pathogenicity feature identification	SVM	MSA, phylogenetic tree construction [20,21]
	Alignment free biological sequence analysis	Identification of viral genomes	RNN	BLAST, Sequence clustering, HHPRED [27]
			CNN	BLAST [28]
protein structure analysis	Post translational modifications	Phosphorylation sites prediction	KNN	Local sequence similarity [53]
			CNN	K-mer based clustering (CD-HIT), BLAST [55]
		Glycosylation sites prediction	ensemble SVM	curated glycosylated protein database (O-GLYCBASE) [54]
	Protein structure prediction	Protein contact prediction	CNN	MSA [72]
		Prediction of distances between pairs of residues	CNN	MSA, HHPRED, PSI-BLAST [77]
systems biology	inference of biological networks	Gene regulatory network prediction	SVM	GeneNetWeaver, RegulonDB [81]
		Protein-protein interaction network prediction	SVM	Domain affinity and frequency tables [90]
			Elastic-net regression	Protein descriptors [91]
	Analysis of biological networks	Drug target prediction	K-means	Network analysis tools [98]
		Drug side effect prediction	SVM	Genome scale metabolic modeling [112]
		Drug Synergism prediction	Random Forest Ensemble	A chemical-genetic interaction matrix [117]
	Multi-omics integration	Cancer subtype prediction	Neighborhood based clustering	Similarity based integration [141]
		Drug response prediction	logistic regression	Cancer hallmarks datasets, pathway data [144]
biomarker analysis for disease research	Disease-associated genes investigation	Pulmonary sarcoidosis genes identification	Hierarchical clustering	Differential expression analysis [150]
		Identification of miRNA-disease association	NMF	Disease semantic information and miRNA functional information [151]
		Disease-phenotype visualization	t-SNE	OMIM database and human disease networks [154]
	Biomarker discovery	Cancer diagnosis	SVM	Reference gene selection [170]
		Biomarker signature identification	SVM	Network-based gene selection [167]
		Cancer outcome prediction	Random forest	Evolutionary conservation estimation [181]

## Data Availability

Not applicable for a review article.

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
