# Peer review of "Incorporating Machine Learning into Established Bioinformatics Frameworks"

_ijms, 2021, doi:10.3390/ijms22062903_

Round 1

Reviewer 1 Report

In this study, Auslander et al. proposed a comprehensive review of using machine learning in bioinformatics studies from sequencing data, gene expression, or SNP data. The idea is of interest and this study holds the potential for publication. I have some major comments to help to improve the study:

1. The title looks general. It is better if the authors could make it more specialized since machine learning in bioinformatics is very common.

2. Sometimes the authors used "biomedical research", but I think it is better if the authors use "bioinformatics research" to match with this specific review.

3. The authors mentioned deep learning, but they did not review any bioinformatics paper that used deep learning. They just list deep learning as a challenge and future, but deep learning is already used a lot recently in bioinformatics. If the authors aimed to target the review into machine learning only, I think don't need to discuss the challenges of deep learning.

4. NLP can be treated as a machine learning technique and has been used a lot in bioinformatics, thus the authors should have some discussions on these models.

5. Some latest machine learning-based bioinformatics works for sequence analysis should be mentioned, i.e., PMID: 33036150 and PMID: 33260643.

6. It is better if there are some tables to show the differences of representative works in each section.

Reviewer 2 Report

The article reviews machine learning frameworks for bioinformatics. 

It is well written and well organized. And it is actually a very useful guide for finding the most suitable algorithms for a given biological problem. 

The article is well organized, my only concern is related to Figure 1. This taxonomy is useful to summarize the main machine learning algorithms and the classes they belong to. However, I would have expected to find this structure in the text, or at least some references. But it seems that the text in the article is not organized according  to this taxonomy. For example there are no references to k-means. I would suggest to keep in the text the structure used in the figure, so as to make the reading easier. 

Another point I want to highlight, among the dimensionality reduction techniques that are widely used in omics analysis there are non-negative matrix factorizations.  There are different tools that use NMF for biological analysis (e.g. https://doi.org/10.1186/1751-0473-8-10 , https://doi.org/10.1016/j.gpb.2013.06.001 , https://doi.org/10.1093/nar/gkn335 , https://doi.org/10.3390/app9245552). I suggest to add this further category.

Round 2

Reviewer 1 Report

The authors have addressed all of my previous concerns. One minor thing: I have found that the authors mentioned "These applications include identification of promoters [36,37], enhancers [38,39] ...". As there are some latest papers that also used deep learning in identifying promoters (PMID: 31750297) and enhancer (PMID: 33539511), it is necessary to include them also in this literature review. Especially, it is a review paper and must cover a comprehensive review.

Author Response

The only comment of reviewer 1, after the previous revision, was that we needed to cite two additional references. This has been done, the references in question are:

Ref. 36 (ll. 505-507)

Ref. 38 (ll. 510-511)